# A 4-Gene Signature of CDKN1, FDXR, SESN1 and PCNA Radiation Biomarkers for Prediction of Patient Radiosensitivity

**DOI:** 10.3390/ijms221910607

**Published:** 2021-09-30

**Authors:** Orla Howe, Lisa White, Daniel Cullen, Grainne O’Brien, Laura Shields, Jane Bryant, Emma Noone, Shirley Bradshaw, Marie Finn, Mary Dunne, Aoife M. Shannon, John Armstrong, Brendan McClean, Aidan Meade, Christophe Badie, Fiona M. Lyng

**Affiliations:** 1School of Biological & Health Sciences, Technological University Dublin, Grangegorman, D07 XT95 Dublin, Ireland; lisa.white@almacgroup.com; 2Radiation & Environmental Science Centre, FOCAS Research Institute, Technological University Dublin, Camden Row, D08 CKP1 Dublin, Ireland; d13123391@mytudublin.ie (D.C.); radmorejane@gmail.com (J.B.); aidan.meade@tudublin.ie (A.M.); fiona.lyng@tudublin.ie (F.M.L.); 3School of Physics & Clinical & Optometric Sciences, Technological University Dublin, Grangegorman, D07 XT95 Dublin, Ireland; 4Centre for Radiation, Chemical and Environmental Hazards, Radiation Effects Department, Public Health England, Oxfordshire OX11 0RQ, UK; grainne.obrien@phe.gov.uk (G.O.); christophe.badie@phe.gov.uk (C.B.); 5Department of Medical Physics, Saint Luke’s Radiation Oncology Network, D06 HH36 Dublin, Ireland; laura.shields@slh.ie (L.S.); brendan.mcclean@slh.ie (B.M.); 6Clinical Trials Unit, Saint Luke’s Radiation Oncology Network at St Luke’s Hospital, D06 HH36 Dublin, Ireland; emma.noone@slh.ie (E.N.); shirley.bradshaw@slh.ie (S.B.); marie.finn@slh.ie (M.F.); mary.dunne@slh.ie (M.D.); 7Cancer Trials Ireland, D11 KXN4 Dublin, Ireland; aoife.shannon@cancertrials.ie (A.M.S.); john.arnstrong@slh.ie (J.A.); 8Saint Luke’s Radiation Oncology Network at St Luke’s Hospital, D06 HH36 Dublin, Ireland

**Keywords:** radiosensitivity, biomarkers, gene expression

## Abstract

The quest for the discovery and validation of radiosensitivity biomarkers is ongoing and while conventional bioassays are well established as biomarkers, molecular advances have unveiled new emerging biomarkers. Herein, we present the validation of a new 4-gene signature panel of CDKN1, FDXR, SESN1 and PCNA previously reported to be radiation-responsive genes, using the conventional G2 chromosomal radiosensitivity assay. Radiation-induced G2 chromosomal radiosensitivity at 0.05 Gy and 0.5 Gy IR is presented for a healthy control (*n* = 45) and a prostate cancer (*n* = 14) donor cohort. For the prostate cancer cohort, data from two sampling time points (baseline and Androgen Deprivation Therapy (ADT)) is provided, and a significant difference (*p* > 0.001) between 0.05 Gy and 0.5 Gy was evident for all donor cohorts. Selected donor samples from each cohort also exposed to 0.05 Gy and 0.5 Gy IR were analysed for relative gene expression of the 4-gene signature. In the healthy donor cohort, there was a significant difference in gene expression between IR dose for CDKN1, FXDR and SESN1 but not PCNA and no significant difference found between all prostate cancer donors, unless they were classified as radiation-induced G2 chromosomal radiosensitive. Interestingly, ADT had an effect on radiation response for some donors highlighting intra-individual heterogeneity of prostate cancer donors.

## 1. Introduction

In the last 10+ years, multidisciplinary research teams created within the European Union have focused their efforts on biomarkers and bioassays to predict radiation -sensitivity and -susceptibility in patient cohorts. These include the Multidisciplinary European Low Dose Initiative (MELODI) platform linked to other European platforms such as Low Dose Research towards Multidisciplinary Integration-DoReMi (2010–2015) [1] and Releasing the European Network in Biodosimetry -RENEB (2012–2015) [2]. These European networks reviewed the different Biomarkers used for radiation epidemiology studies [3,4], as well as screening assays for individual radiation sensitivity and susceptibility [5]. Two significant RENEB gene expression studies explored microarray and quantitative real-time PCR (qRT-PCR) gene expression platforms for biological dosimetry in donor blood samples through two [6] and five [7] established European gene expression laboratories. Both studies demonstrated accurate dose estimates through qRT-PCR that focused on specific radiation responsive genes.

The molecular mechanisms of cellular response to high and low doses of ionising radiation are well known and linked to the DNA damage response (DDR). It is therefore not surprising that many gene expression studies have focused on genes central to the DDR such as the ATM/CHK2/P53 pathway signaled by IR-induced double strand DNA damage. The expression of 4 genes CCNB1, CDKN1A, BBC3 and GADD45 in the DDR was studied in the lymphocytes of breast cancer patients with differences in sensitivity to radiation treatment, however only CDKN1A gene expression demonstrated discrimination between normal and severe reactions in patients (up to 91%) [8]. CDKN1A is a cyclin-dependent kinase inhibitor-1A (also known as p21) that codes for a protein that binds and inhibits CDK1, CDK2 and CDK4/6 complexes to regulate or halt cell cycle progression at G1 and S phase. This activity is regulated by p53 via ATM and CHK2 to regulate the cell cycle checkpoints in response to IR, but it is also known to be regulated by P53 independent pathways to promote the assembly of these complexes particularly in the G2/M phase of the cell cycle [9]. A study by Manning et al., 2013 focused on multiplex qRT-PCR (MQRT-PCR) of 13 genes transcriptionally regulated by P53 in blood samples irradiated with 0.1–4 Gy X-rays at 2 and 4 h post exposure. From the low dose estimation curves, three genes FDXR, DDB2 and CCNG1 gave the best dose estimates, with the majority of the genes showing a higher level of expression at 24 h than 2 h post-exposure, with the exception of CDKN1A, PCNA and MYC genes [10]. Ferredoxin reductase (FDXR) has emerged in many of the above studies as a highly radiation responsive gene in both ex vivo and in vivo blood samples from radiotherapy patients including those reported in the RENEB studies [6,7,8]. More recently, a 28-fold change of FDXR gene expression was reported in blood samples that were irradiated with 2 Gy X-irradiation ex vivo and the PMBCs used for Nanopore sequencing analysis [11]. FDXR is a mitochondrial membrane-associated flavoprotein required for biogenesis of iron–sulfur clusters and for steroidogenesis. FDXR transfers electrons from NADPH to mitochondrial cytochrome c and is regulated by P53 to function in iron homeostasis. It is also involved in apoptosis signaling through reactive oxygen species (ROS) [12].

Although CDKN1A and FDXR were established as highly radiation-responsive genes, many additional genes that were analysed by MQRT-PCR were also deemed to be suitable for radiation dosimetry in response to IR. In a study of 108 samples that were analysed by both DNA microarray (216 arrays) and MQRT-PCR in both whole blood and lymphocyte samples, 570 genes were upregulated in blood samples compared to 232 genes upregulated in lymphocytes in the microarray analysis, with a significantly high number of genes downregulated in lymphocytes compared to whole blood. Among these genes, SESN1 (Sestrin 1) was one of 9 genes that was consistently upregulated in both whole blood and lymphocytes, and PCNA1 (proliferating cell nuclear antigen 1) was specifically upregulated in whole blood with both genes deemed to be potential biomarkers of radiation response [13,14].

SESN1 first identified in 1994 [15] is a p53-response gene (solely activated by p53) which functions to regenerate over oxidized peroxiredoxins in response to IR, among other members of the sestrin family such as SESN2 [16]. SESN1 and SESN2 can negatively regulate mTOR and its signaling, in which mTOR functions to promote tumorigenesis through proliferation and chemoresistance. Therefore, SESN1 and SESN2 are important IR-induced mediators of the DDR via P53 and mTOR signaling pathway [17]. PCNA1 also has a direct role in the DDR. It was first reported to act as a processivity factor of DNA polymerase δ in eukaryotic cells [18,19] and is placed as a clamp at replication forks to coordinate DNA replication and DNA repair. It also acts as a loading clamp for additional DDR proteins to recruit to the site of DNA damage and bind [20]. The interaction of CDKN1 (p21) with PCNA is also of interest because it facilitates the inhibition of DNA synthesis in response to IR, promoting cell cycle checkpoint arrest until DNA repair is complete and therefore promoting genetic stability [21]. The interaction of CDKN1 and PCNA1 causes G2/M cell cycle arrest by blocking interaction of Cdc25C with PCNA1 [22]. CDKN1 and PCNA1 were also reported to interact with p300 altering Histone acetylase activity (HAT) in Nucelotide Excision Repair (NER) processes of UV irradiated cells [23].

It is evident from the literature that CDKN1, FXDR, SESN1 and PCNA1 are highly radiation responsive genes for radiation dosimetry and as demonstrated in the numerous donor blood studies published to date. We therefore specifically targeted these genes as a 4-gene signature for MQRT-PCR analysis. Whole blood samples from a cohort of prostate cancer patients and a cohort of healthy donor controls were first characterised by the G2 chromosomal radiosensitivity assay. This bioassay first described in irradiated cells by Parshad and colleagues [24,25] is now a well-known cytogenetic assay and therefore a biomarker for cellular radiosensitivity [26]. The assay is well-established and optimised at our own laboratory for predicting cellular radiosensitivity in blood samples obtained from many individuals in various cancer and healthy cohorts [27,28,29,30,31]. Herein, we report the potential of CDKN1, FDXR, SESN1 and PCNA packaged as a 4-gene signature as a potential radiosensitivity biomarker and validated using the G2 chromosomal radiosensitivity profile of individual samples of both the prostate cancer and healthy donor cohort exposed to low doses of IR.

## 2. Results

### 2.1. G2 Chromosomal Radiosensitivity

#### 2.1.1. Healthy Control Donors

The radiation-induced G2 chromosomal radiosensitivity scores were obtained from blood samples of 45 healthy control donors exposed to 0.05 Gy and 0.5 Gy IR doses in culture. These scores ranged from 8–68 and 92–210 aberrations/100 metaphases (abs/100 meta), respectively, with a higher sensitivity range evident at 0.5 Gy compared to the lower 0.05 Gy dose. In contrast, lower relative variability expressed as the coefficient of variation (CV) was less for 0.5 Gy (22.3 abs/100 meta) when compared to 0.05 Gy (37.9 abs/100). Table 1 displays the statistics for this cohort of healthy control donors at both IR doses. The 90th percentile value was calculated for both doses as this has been reported in numerous G2 radiosensitivity studies incorporating healthy control cohorts as a cut-off value for G2 radiosensitivity for all prostate donor cohorts for the same study [27,29,30,31,32]. Therefore, all donors (healthy control and prostate cancer) exceeding 50 abs/100 meta for 0.05 Gy and 152 abs/100 meta for 0.5 Gy were deemed to be radiosensitive. The difference between the mean radiation-induced G2 scores for 0.05 Gy compared to 0.5 Gy in this cohort were highly significant (two-tailed *p* value < 0.0001) by the Mann–Whitney test and this is also illustrated in Figure 1 for both doses in the healthy control donors. Similarly, when G2 scores for each dose were matched per donor, the difference was highly significant (two-tailed *p* value < 0.0001) by Wilcoxin signed ranks test.

#### 2.1.2. Prostate Cancer Donors

Radiation-induced G2 chromosomal radiosensitiviy scores were also derived from 12 prostate cancer donors exposed to 0.05 Gy and 0.5 Gy doses in culture and which were sampled on two separate visits to the hospital. At visit 1 the baseline donor sample was taken post-diagnosis with prostate cancer once the study eligibility criteria were met. At visit 2, the same prostate patients had undergone androgen deprivation therapy (ADT) prior to their radiotherapy treatment regime. At 0.05 Gy IR, the mean radiation-induced G2 scores were only slightly higher in the prostate cohort at baseline (37 abs/100 meta) and ADT (38.8 abs/100 meta) compared to the healthy control cohort (33.7 abs/100 meta), and all mean values fall below the 90th percentile radiosensitive cut-off value of 50 abs/100 meta. In contrast, for 0.5 Gy the mean radiation-induced G2 scores exceeded the 90th percentile radiosensitivity cut-off of 152 abs/100 meta for both baseline (167.5 abs/100 meta) and ADT (162.6) compared to the healthy control cohort (122.6 abs/100 meta). It is well reported in the literature that 0.5 Gy IR is the most radiosensitive dose at G2 phase of the cell cycle due to checkpoint inefficacy [33,34], and prostate cancer patients exhibit elevated G2 radiosensitivity compared to healthy control donors at this dose [28,30,32].

Figure 1 below exhibits the differences of the mean radiation-induced G2 scores for the baseline and ADT prostate cancer cohort compared to the healthy control cohort for both 0.05 Gy and 0.5 Gy dose. For both baseline and ADT prostate data, they were significantly different when dose was compared (two-tailed *p* > 0.0001) by the Mann–Whitney test and the Wilcoxin signed ranks test (two-tailed *p* > 0.0005).

#### 2.1.3. G2 Chromosomal Radiosensitivity Shifts in Prostate Cancer Donors

As previously described, the data for 0.5 Gy in both prostate cancer sub-sets (baseline and ADT) were elevated and exceeded the 90th percentile radiosensitivity cut-off value (152 abs/100 meta) and was analysed more closely to determine if there had been any change to radiation-induced G2 chromosomal radiosensitivity between the two visits at baseline and later at ADT. At baseline, 8 out of 12 samples were G2 radiosensitive exceeding the cut-off value compared to 5 out of 12 donors after ADT. However, interestingly, when each donor was directly compared from baseline to ADT using the radiosensitivity cut-off value =/- 5%, only 50% (6 out of 12 donors) retained the same G2 radiosensitivity status. Figure 2 below illustrates the G2 radiosensitivity changes from baseline to ADT for each of the 12 prostate cancer donors. The G2 chromosomal radiosensitivity reduced in PC2-4 and PC10 and increased in PC8 and PC11. Statistically, the pairing of G2 scores for baseline vs. ADT per prostate cancer donor was not significant (two-tailed *p* = 0.7910) by Wilcoxin matched pairs signed ranks test. Based on these observations, it can only be assumed that the hormone changes from the androgen deprivation therapy had an effect on G2 chromosomal radiosensitivity for some but not all patients, adding to the complexity of intra-individual heterogeneity to radiation response.

### 2.2. Relative Gene Expression of 4-Gene Panel

#### 2.2.1. Healthy Control Donors

Multiplex quantitative real-time PCR (MQRT-PCR) was carried for the 4 target genes CDKN1A, FDXR, SESN1 and PCNA against a housekeeping control Hypoxanthine-Guanine phosphoribosyl transferase 1 (HPRT1) and the 0 Gy samples at the laboratory of Public Health England in Oxfordshire UK. Therefore, not all healthy control and prostate cancer donor samples from the G2 chromosomal radiosensitivity study above were available at the time of the research visit. A selection of 14 healthy control samples were tested for each gene target and the Ct (cycle threshold) values were normalized to the HPRT1 housekeeping control and the log2 (fold change) values statistically analysed for both 0.05 Gy and 0.5 Gy IR doses. The fold changes of all genes compared together at 0.05 Gy and then at 0.5 Gy were not significantly different with *p* = 0.9294 and *p* = 0.0768, respectively, determined by the Kruskal–Wallis test. However, the Mann–Whitney test for the mean differences of 0.05 Gy compared to 0.5 Gy for each gene determined that CDKN1 (*p* = 0.0496), FDXR (*p* = 0.0497) and SESN1 (*p* = 0.0021) were significantly different and compared to PCNA (*p* = 0.0767) which was not significant. On closer inspection of fold change gene expression values illustrated in Figure 3 below, the 3 genes CDKN1A, FDXR and SESN1 show higher gene expression than PCNA particularly for 0.5 Gy compared to 0.05 Gy in a dose-dependent response.

#### 2.2.2. Prostate Cancer Donors

A selection of 7 prostate cancer donor samples at baseline with 3 of these prostate patients who had completed Androgen Deprivation Therapy (ADT) prior to their radiotherapy treatment were available at the time the MQRT-PCR work was carried out as above in the UK, and are presented in Figure 4 and Figure 5 below. As for the healthy control samples, the fold change values for the 4 normalised target genes CDKN1A, FDXR, SESN1 and PCNA were statistically analysed for the 7 baseline samples at 0.05 Gy and 0.5 Gy by the Kruskal–Wallis test indicating no significant difference in gene expression of all samples irrespective of IR dose, with *p* = 0.6455 for 0.05 Gy and 0.417 for 0.5 Gy donor samples. The Mann–Whitney test comparing IR dose of 0.05 Gy with 0.5 Gy for each gene showed no significance for CDKN1 (*p* = 0.7104), FDXR (*p* ≥ 0.9999), SESN1 (*p* = 0.053) and PCNA (*p* = 0.9015). However, on closer inspection of gene expression values, both SESN1 and CDKN1 had higher expression levels for 0.5Gy for most samples. Both Kruskal–Wallis and Mann–Whitney statistics were used to analyse gene expression of 3 genes and then 2 genes to determine the discrimination power of the gene signatures as 4, 3 or 2 genes at 0.5Gy with the inclusion of both SESN1 and CDKN1. Although the *p* values were not statistically significant most likely due to the small sample size, there was a small but observable change in the *p* value. This was observable for 4 genes (*p* = 0.417), 3 genes of SESN1, CDKN1 and FXDR (*p* = 0.752) and 2 genes of SESN1 and CDKN1 (*p* ≥ 0.999) which highlight that more genes added to a gene signature for screening purposes is advantagous.

Similarly for the 3 prostate cancer donor samples following ADT, no significance was found in gene expression for 0.05 Gy (*p* = 0.9098) and 0.5 Gy (*p* = 0.8056) data, nor for any of the 4 genes between the two doses by Mann–Whitney (all *p* ≥ 0.999). However, on closer inspection of the individual prostate cancer baseline samples as presented in Figure 3, fold changes exceeding 1 were deemed to be significant, with CDKN1A, FDXR and SESN1 exceeding this threshold fold change particularly for 0.5 Gy for all 3 genes with PCNA in contrast below the threshold. This trend correlates to the fold change gene expresison data for the healthy control donor samples.

When fold change gene expression values of the 4 target genes were analysed in the prostate cancer samples following Androgen Deprivation Therapy (ADT) (PCT donors) and compared to the baseline samples (PC donors) for both 0.05 Gy and 0.5 Gy as illustrated in Figure 4, the trend is less evident per donor sample. PCNA gene expression has a fold change of 1.33 for 0.05 Gy and 1.5 for 0.5 Gy in PCT1 while the 3 other genes did not exceed the fold change threshold of 1. In contrast PCT3 exceeded this threshold for CDKN1A (1.36); FDXR (1.18) and SESN1 (1.62) but not PCNA (0.99) at 0.5 Gy, while PCT2 did not exceed the threshold for any gene at 0.5 Gy. Interestingly PCT5 was a donor sample taken at visit 2 (ADT) for the same PC5 donor baseline sample and PCT12 was the second ADT sample for the corresponding PC12 donor baseline sample. Contrasting the fold change gene expression data from both donors at baseline and ADT for the two IR doses 0.05 Gy and 0.5 Gy, there is a change in expression levels of all 4 genes not just the 3 genes CDKN1, FDXR and SESN1 which emerged as the most radiation sensitive genes in the panel. Although the prostate donor sample sets are relatively small for the gene expression study, it must be noted from Figure 4 that each prostate cancer donor has an individualized gene expression profile for the 4 genes and that therapeutic interventions with other factors appear to alter that genetic profile.

#### 2.2.3. Biomarker Correlations

From the data presented above both G2 chromosomal radiosensitivity and relative gene expression of the 4-gene panel as conventional and new biomarkers, respectively, demonstrated a general dose dependent response for 0.05 Gy and 0.5 Gy in all donor samples but there was clearly heterogeneity in individual donor responses. However, the 0.5 Gy dose yielded the most radiosensitive response as expected for most donors particularly evident in radiation-induced G2 chromosomal radiosensitivity. Therefore, 0.5 Gy data from both 14 healthy control donors and baseline prostate cancer donors with corresponding biomarker data (G2 chromosomal radiosensitivity and relative gene expression data of the 4-gene panel) were correlated. From the selected healthy control donors, none were deemed to be G2 radiosensitive. The 90th percentile cut-off value of 152 aberrations/100 metaphases (152 abs/100 meta) as derived from the G2 chromosomal radiosensitivity study in 45 healthy control donors was again used as a threshold. Although 2 out of 14 donor samples did reach the 152 aberrations/100 metaphases (152 abs/100 meta), they did not exceed this value to deem them G2 radiosensitive and gene expression could not be directly correlated with G2 radiosensitivity in the healthy control donor cohort. In contrast, 6 prostate cancer donor samples at baseline with all corresponding biomarker data were compared and contrasted. Five out of 6 donors were deemed to be radiosensitive exceeding the 90th percentile cut-off (152 abs/100 meta with only one 1 donor not G2 radiosensitive at 110 abs/100 meta. On analysis of the 4 genes individually for this latter radiosensitive donor cohort, only SESN1 exhibited a fold change above the threshold (1.31 fold change) with the remaining 3 genes below the threshold. For the G2 radiosensitive prostate cancer donors, 60% (3/5) CDKN1A, 40% FDXR, 80% SESN1 and 40% PCNA gene expression exceeded the fold change threshold. The Wilcoxin matched-pairs signed ranks test for both data sets per donor demonstrated significance (*p* = 0.0313) for each gene, although the Spearmans correlation did not find the pairing significant which was expected with two different biological end-point data sets. Additional Bland–Altman plots were performed as an alternative to correlation analysis to determine the overall degree of agreement between the two radiobiological endpoints of G2 radiosensitivity and the gene expression for each of the 4 genes individually at 05 Gy since this IR dose point was used for both radiobiological endpoints. These plots are presented in Appendix A. In total, 95% of the data points fell between the standard deviations of the mean differences. The limits of agreement and mean values were similar for all 4 genes and indicated large variability between the two radiobiological endpoints which was expected. Therefore, while both radiobiological endpoints are useful biomarkers of cytogenetic and molecular response to low dose radiation in donor samples, they cannot be directly correlated.

Similarly, correlations of clinical toxicity scores from prostate cancer donors were performed with G2 scores and gene expression of each individual gene PCNA, FXDR, CDKN12 and SESN1 using the non-parametric Spearmans correlation test. Clinical toxicity scores from both baseline (pre-RT) and post-radiotherapy treatment were correlated but were not significant. While both of the radiobiological endpoints using low doses of IR can discriminate radiation response between normal healthy control and cancer donors, they cannot discriminate between the clinical toxicity grades 1 and 2. The selected test genes may also be related to cancer predisposition rather than radiosensitivity which adds further complexity to the data.

## 3. Discussion

The identification of biomarkers that predict patient response and outcomes prior to radiotherapy treatment will be an important advancement to radiobiology in the future, with large efforts made by European and International Radiation Research platforms. It was recognized at a recent workshop of the National Cancer Institute (NCI), USA that combination biomarkers measuring multiple analytes instead of single biomarkers should be explored in future biomarker studies, and the discovery of predictive markers should be validated in independent patient cohorts [35]. In addition, a meeting of the Association of Radiation Research (ARR) in 2016 recognized that cytogenetic assays are established conventional biomarkers of radiation exposure and ideally should be combined with new emerging biomarkers such as transcriptionally altered genes by radiation [36]. Herein, we present cytogenetic data from an established cytogenetic assay, the radiation-induced G2 chromosomal radiosensitivity assay at 0.05 Gy and 0.5 Gy low doses of ionizing radiation (IR) for healthy control (*n* = 45) and prostate cancer (*n* = 12) donor cohorts (at baseline and Androgen deprivation therapy (ADT)). This cytogenetic assay was used to validate a 4-gene signature of the radiation responsive genes CDKN1A, FDXR, SESN1 and PCNA. This 4-gene signature represents a new emerging transcriptional biomarker for prediction of radiation response and our objective was to compare the relative gene expression of the 4 genes with G2 chromosomal radiosensitivity in the donor cohorts for 0.05 Gy and 0.5 Gy IR.

Peripheral blood lymphocytes (PBL) derived from whole blood samples of all donors were used as it is the preferred source of tissue for assays of normal tissue toxicity (NTT) response/radiosensitivity in patients. This is due to the ease at which samples can be taken from patients that is minimally invasive, and the rapid speed in which data can be generated. The use of PBL in donor cohorts is ideal for the development of a predictive biomarker of radiosensitivity. A recent review on ex vivo-induced biodosimetric markers to predict acute or late radiation toxicity effects in radiotherapy patients described the benefits of cytogenetic assays using PBL to help evaluate individual radiosensitivity according to dose and genetic status, because radiation-induced damage is related to altered DNA repair mechanisms and therefore cellular radiosensiviity [37]. Herein, 4 genes related to the ATM/CHK2/P53 pathway in the DNA Damage Response (DDR) were selected for this study including CDKN1A, FDXR, SESN1 and PCNA possibly directly associated with the underlying mechanisms of G2 chromosomal radiosensitivity in individual donor cohorts. It is evident from our data that all donor samples present an individual radiation dose-dependent response for the two doses 0.05 Gy and 0.5 Gy and was statistically validated. The 0.5 Gy (deemed the most radiosensitive dose) exhibited a more enhanced G2 chromosomal radiosensitive response universally in all healthy control and prostate cancer donors (at baseline) when using the 90th percentile cut-off value of 152 aberrations/100 metaphases calculated from the healthy control donor cohort (*n* = 45).

However, when G2 chromosomal radiosensitivity scores from prostate cancer donors (*n* = 12) were compared at 0.5 Gy baseline with androgen deprivation therapy (ADT) only 50% retained their G2 radiosensitivity status indicating that the hormonal effects of ADT may be contributory factors for radiosensitivity. ADT is the foundational treatment for men with advanced prostate cancer as it reduces circulating levels on androgens, and it is given prior to and in combination with radiotherapy treatment [38]. ADT and radiotherapy are known to have synergistic effects but different cellular mechanisms. Androgens are a class of lipophilic steroid molecules that cross the cell membrane and bind to cytosolic androgen receptors, becoming active and binding to DNA to transcriptionally alter genes involved in cell survival and growth. This pathway is blocked by ADT so that cell death occurs instead and in synergy, radiotherapy induces cell death with unrepairable DNA double strand breaks [39]. In addition, the activated androgen receptor has been reported to help repair radiation-induced double strand DNA breaks which can upregulate androgen receptor signaling [40]. Given the different but synergistic signaling mechanisms of ADT with radiation, it was not surprising to record a change in G2 chromosomal radiosensitivity at ADT from some of the prostate cancer donor baselines samples. This does however highlight that for any predictive biomarker used in the clinic, regular sampling at various stages of the radiotherapeutic regime would be more advantageous for prediction, prognosis and therapeutic outcome.

For relative expression of the 4-gene signature in all donor cohorts, it was evident that CDKN1A, FDXR and SESN1 were the most radiation-responsive genes at both 0.05 Gy and 0.5 Gy doses for both the healthy control and prostate (baseline) donor cohorts. In contrast, PCNA was not as radiation-responsive for either dose in the prostate cancer donor cohort but was expressed over 1 fold change at 0.5 Gy for some of the healthy control donors. Interestingly, for one of the ADT donors (PCT5), PCNA was the most radiation-responsive gene compared to the other 3 genes, and this transcriptional response was completely different to the baseline response (PC5) for the same donor, again highlighting a transcriptional alteration due to the underlying hormonal response. For this study, the ADT sample set (*n* = 3) was too small to further observe the role of radiation-responsiveness of PCNA or the transcriptional changes in the other 3 target genes induced by ADT. In an attempt to correlate G2 chromosomal radiosensitivity with gene expression of the 4 individual genes from the 4-gene signature, a sub-set of healthy control and prostate (baseline) donor samples with both sets of available data at 0.5 Gy were selected. Donor samples deemed to be G2 chromosomal radiosensitive (152 abs/100 meta) were our focus, because although CDKN1A, FDXR and SESN1 were clearly radiation-responsive genes, this was consistently observed in both healthy control and prostate cancer donors. From the selected healthy control donors (*n* = 14) none were deemed to be G2 radiosensitive and from the prostate cancer donors (*n* = 6), 5 were deemed to be G2 radiosensitive, so direct correlations of gene expression with G2 radiosensitivity could not be performed with the healthy control donors. For the prostate cancer donors (at baseline) the gene expression was as follows: SESN1 (80%) > CDKN1A (60%) > FDXR (40%) > PCNA (40%) indicating that all genes in the 4-gene signature were related to G2 radiosensitivity, however statistical significance was not found in a correlation analysis of G2 scores with each gene. When gene expression of the 4 genes were compared for the doses used (0.05 and 0.5Gy), there was no statistical significance, but a change in *p* value was evident for 4 genes compared to 3 and 2 genes indicating the greater discriminating power of a 4-gene signature. However, this cohort was relatively small and would benefit from a larger donor sample size and from radiosensitive cancer patients other than prostate cancer for further validation of the 4-gene signature. Further statistical correlations of G2 scores and gene expression of each of the 4 genes was performed with the clinical toxicity scores of the prostate cancer donors at both baseline (pre-RT) and post-RT but there was no significance with the clinical toxicity scores of 1 and 2. Therefore, while the data presented within shows a radiobiological response to low doses of IR (0.05 Gy and 0.5 Gy) with both endpoints and can discriminate between health control and cancer cohorts, it cannot discriminate between the low clinical toxicity scores 1 and 2. This data is therefore indicative of cellular and molecular radiobiological responses with insights into cancer predisposition but is not indicative of clinical radiosensitivity. Further large scale patient studies to align both cellular and clinical radiosensitivity are needed to find a potential biomarker for clinical purpose.

In the quest for biomarker discovery and validation, our data highlights several important points. (1) Intra-individual heterogeneity in radiation response reported in numerous biomarker discovery studies can determine the reproducibility of a biomarker assay. Herein, we observed the effect of androgen deprivation therapy (ADT) on G2 chromosomal radiosensitivity along with the expression of the 4 genes compared to the same prostate cancer donors at baseline. This suggests a hormonal effect on radiation response, which appears reasonable considering the synergistic mechanisms described above. Intra-individual response specifically in G2 chromosomal radiosensitivity has also previously been reported by many groups including ours without any obvious contributory factors like ADT in this study [31,41,42]. These studies suggest that multiple sampling from the same donor should be considered, although this is often not practical for patient donors. For biodosimetry purposes, inter- and intra- laboratory comparisons for gene expression on PBL was also considered and validated by the RENEB studies and the NATO biodosimetry laboratory inter-compariosn study [2,7,43]. Because the reproducibility of the biomarker assay can be determined by intra-individual heterogeneity, it is essential to put measures in place to ensure a reliable robust assay of radiosensitivity. From our study, we propose (a) the inclusion of a healthy control donor cohort alongside patient cohorts. (b) multiple samples per donor if practicable and at various stages of radiotherapy treatment to detect changes in radiosensitivity status, (c) application of at least 2 dose radiation points (including the sham-irradiated 0 Gy control) to establish dose-dependent radiation responsiveness and finally (d) establishing radiosensitve and radioresistant sub-groups within cohorts aligned to any available clinical data.

(2) The radiosensitivity phenotype is compounded by diverse underlying genetic factors that go beyond DNA damage and repair processes, and may include inherited mutations or single-nucleotide polymorphisms (SNP) of genes as well as possible transcription variants. A recent genome-wide SNP analysis with CDKN1A gene expression in different healthy cohorts (including dizygotic and monozygotic twins) reported genetic variation accounts for 66% transcriptional response to radiation. Furthermore, SNPs that were located in CDKN1A transcription factor genes *ETV6* and *KLF7* as well as other genes such as *RPA3* and *AKIP* [44]. A recent study that characterized 14 FDXR transcript variants that were radiation- induced in vivo in patients but not detectable at basal level indicates that measuring radiation-induced alternative splicing may be an important feature of patient radiosensitivity [45]. Understandably, while there has been much focus on genes of the DDR and DNA repair mechanisms, genes from other related biological processes have shown great promise as biomarkers of radiation response and susceptibility to radiation-induced toxicity. For example, genes that play a role in the inflammatory response such as ARG1, BCL2L1 (upregulated) and MYC (downregulated) were found in blood of patients undergoing radiotherapy treatment for endometrial or head and neck cancers [46]. Similar to the gene expression data presented within, the majority of transcriptomic studies to date are on mRNA that subsequently codes for a cellular protein, estimated to constitute only 2% of the transcriptome. The remaining 98% of the transcriptome is non-coding RNA that was assumed for a long time to be non-functional. However, in recent years, transcriptomic analysis have evolved to include the miRNA (miR) transcriptome studies by microarrays [47,48] or next generation sequencing [49,50] to identify their role in radiation response. miRNAs are small non-coding sequences of 19–21 nucleotides and are important modulators of gene expression in cell cycle control, DNA repair and apoptosis. We recently reported miR 152-3p, miR25-5p and 92-15p that were modulators of PTEN and CCDN1 as potential biomarkers of radiosensitivity correlating with G2 chromosomal radiosensitivity in Ataxia Telangectasia cells [51]. After miR analysis, RT-PCR analysis of mRNA established their significance as radiation responsive genes and interestingly are both involved in cell cycle regulation aligned to the DDR. The role of miR’s as potential biomarkers were extensively reviewed by Labbe and colleagues [52] although they highlighted the fact that the majority of studies are in prostate cancer cell lines with more research required in patient samples. Given the known stability of miR’s in biofluids [53,54,55] and the fact that mRNA and miRNA (among other non-coding RNA subtypes) can be extracted from the same donor samples, the future of transcriptomic biomarker discovery will most likely involve other types of RNA.

(3) Combination biomarkers measuring multiple analytes instead of single biomarkers is required in future studies to advance biomarker discovery. It is evident that cellular radiosensivitity is a complex phenotype controlled by genes in diverse cellular signaling mechanisms including cell cycle regulation, DNA damage response (DDR) and DNA repair, apoptosis and carcinogenesis. Inter-individual heterogeneity in radiosensiivity is due to a combination of these genetic factors adding to the complexity. While omics approaches can determine the contributory genetic determinants, other established radiobiological assays can determine radiation response. However, the discriminatory power of these assays are too low for use as stand-alone radiosensitivity biomarkers in the clinic. As suggested by the National Cancer Institute [35] combination biomarkers measuring multiple analytes should be used in future biomarker studies and as presented within. Both mRNA and chromosomes (irradiated in culture at 0.05 Gy and 0.5 Gy IR) were extracted from the same donor samples in all cohorts and used for G2 chromosomal radiosensitivity and gene expression, respectively, as different but complimentary biomarker approaches. This multiparametric approach, combined into an entire prognostic profile, might provide better discrimination for the discovery of biomarkers to eventually be adopted into clinical practice [37].

## 4. Materials and Methods

### 4.1. Blood Samples and Donors

Peripheral blood samples were obtained from 45 healthy control donors (age range 21–64 years old) and 12 prostate cancer patients (age range 57–85 years old). It is well know that age is a risk factor for prostate cancer [56] and although donor samples could not be directly matched according to age, 50% of the prostate cancer donors were ≤65 years of age. Ethical approval was obtained by TUDublin ethics committee and St Lukes hospital in 2012. The prostate cancer patients for this study were recruited from the Cancer Trials Ireland (formerly All Ireland Cooperative Oncology Research Group, ICORG) trial 08-17 which is entitled “A Prospective Phase II Dose Escalation Study Using intensity modulated radiotherapy (IMRT) for High Risk N0 M0 Prostate Cancer (NCT00951535)”. The primary endpoint is to determine if dose escalation up to 81 Gy using IMRT for high risk localised prostate cancer can provide prostate specific antigen (PSA) relapse-free survival similar to that previously reported. All patients were prescribed either six months or three years of neo-adjuvant/adjuvant hormone therapy using non-steroidal anti androgens (NSAA) and luteinizing hormone releasing hormone (LHRH). Clinical details of the prostate cancer patients at baseline are provided in the Appendix A) and include Gleason score, TNM and PSA levels prior to hormone and RT treatment. After the neoadjuvant hormone therapy, patients were treated with radiation therapy, where dose escalation was allowed up to a maximum of 81 Gy from a baseline 75.6 Gy, with treatment delivered by intensity modulated radiotherapy. PSA levels and gastrointestinal and genitourinary (GI/GU) toxicities were recorded prior to treatment, during treatment and at follow up using the National Cancer Institute Common Terminology Criteria for Adverse Events (NCI CTCAE) grading system, version 3. Patients were followed up regularly at two months post radiation therapy (RT), eight months post RT, six-monthly thereafter until Year 5, and annually thereafter until Year 9. Our translational research study was approved by the St Luke’s Radiation Oncology Network Research Ethics Committee and all research was performed in accordance with relevant guidelines and regulations. Informed consent was obtained from all participants. Fresh whole blood was drawn into Lithium-heparin tubes at St. Luke’s Radiation Oncology Network, Dublin, and were coded before being transferred to the Technological University (TU) Dublin laboratory.

### 4.2. Whole Blood Cultures and Irradiation

Whole blood cultures were set up in duplicate flasks per dose (0, 0.05 and 0.5 Gy) using 2 mL blood in 18 mL supplemented RPMI-1640 media with 0.2 mL mitogen (45 mg Phytohaemagglutinin (PHA) PAA Laboratories) to stimulate cells into a cell cycle. Cultures were irradiated after 72 h with an x-ray linear accelerator (LINAC) at SLRON SLH, Dublin. The low doses of 0.05Gy and 0.5Gy were selected on the basis of previous studies performed at our Institute. Both doses show differential gene expression for direct and bystander radiobiological effects [57] while 0.5 Gy IR is well reported as the most radiosensitive dose for the G2 chromosomal radiosensitivity assay [24,25,26,27,28,29,30,31,58,59]. The dose rate was approximately 1.5 Gy/min during these experiments and was determined from a distance corrected measurement of the in-beam axial dose at an 80 cm source to chamber distance. This was measured using a secondary standard ionization chamber within a water equivalent phantom. The LINAC was calibrated in accordance with the 1990 IPSM code of practice [60] by the Medical Physics Department at SLRON SLH, such that 100 Monitor Units (MU, a measure of “beam on” time) delivered a dose of 0.1 Gy at 1.4 cm deep in water positioned 100 cm from the source for a 10 × 10 cm^2^ field. In order to achieve a uniform irradiation of flasks in practice, the irradiation conditions were altered from those at calibration. A 30 × 35 cm^2^ field was used to deliver each dose. The flasks were also positioned 10 cm deep in a water equivalent phantom 90 cm from the source. At 90 cm from the source 100 MU delivers a dose of 0.0812 Gy at 10 cm deep in water for a 10 × 10 cm^2^ field. The number of MU required to deliver each of the doses outlined above must be corrected for the different scatter conditions present with the larger field size (30 × 35 cm^2^). A correction factor of 1.1372 was therefore applied, which is the ratio of the field area of a large field to a smaller one. Thus at 90 cm from the source, 100 MU delivers a dose of 92.34cGy (81.2·1.1372), and so the delivery of 0.05 Gy required 6 MU and 0.5 Gy required 55 MU. The calculated doses were verified using Gafchromic EBT3 film (Ashland Inc., NJ, USA). The film was calibrated against a Farmer type ionization chamber using the triple channel dosimetry method [61]. The film was scanned using the single scan protocol [62] on an Epson Expression 10000 XL scanner using the recommended scanning resolution of 72 dpi in a 48-bit RGB format [60,61,62,63]. Glass was placed over the calibration and test film during scanning to minimize ringing artefacts. The film was analysed using FilmQA Pro (Ashland Inc.).

### 4.3. G2 Chromosomal Radiosensitivity Assay

Thirty mins post irradiation all blood cultures were exposed to 200 μL of colcemid (1 μg/mL stock) (Gibco) at 37 °C with 5% CO_2_ for 60 min to arrest cells in metaphase. Chromosomes were then harvested from the blood cultures and slides prepared and stained with 2% Giemsa solution as previously described in detail [27,29,30,31]. G2 chromosomal radiosensitivity scores were the total number of structural aberrations (chromatid breaks and gaps) per 100 metaphases microscopically analysed for each donor per dose (0 Gy, 0.05 Gy and 0.5 Gy). Radation-induced G2 chromosomal radiosensitivity scores were calculated by subtracting the G2 score of the 0 Gy control (non- irradiated spontaneous aberrations) from the irradiated (0.05 Gy or 0.5 Gy) G2 scores per donor.

### 4.4. RNA Isolation and cDNA Synthesis

At 1 h post irradiation blood cultures were centrifuged at 300× *g* for 5 min and cell pellets were resuspended in 1 mL of TriReagent (Merck, NJ, USA) (supplemented with 3N Acetic Acid) for the well-known RNA isolation method of Chomczynski and Sacchi (1987). RNA pellets resuspended in 30 µL of RNase free water (Ambion) were quantified on a Nanodrop spectrophotometer and 1 µg of RNA was qualified on 1.2% agarose (Sigma) gels. Gel electrophoresis was set up for 30 min at 80 volts and gels were subsequently visualized using the SynGene G: BOX chemi X6 transilluminator system. Only samples with intact 18 s and 28 s RNA bands were used for cDNA synthesis and MQRT-PCR analysis. cDNA was synthesized by reverse transcription reactions using a High Capacity cDNA Reverse transcription kit (Applied Bio- systems, Foster City, CA, USA) according to the manufacturer’s protocol with 700 ng of total RNA and as previously described [13,14].

### 4.5. Multiplex Quantitative Real Time PCR (MQRT-PCR)

Real-time PCR reactions were run on a Rotor-Gene Q (Qiagen, Hilden, Germany), in triplicate using PerfeCTa^®^ MultiPlex qPCR SuperMix (Quanta Biosciences, Inc. Gaithersburg, MD, USA) with primer and probe sets for each of the target genes at 300 nM concentration each and 2.5 μL of cDNA in 30 μL reaction volume. 3′6-Carboxy fluorescein (FAM), 6-Hexachloro fluorescein (HEX), Texas Red and CY5 (Eurogentec Ltd., Fawley, Hampshire, UK) were used as fluorochrome reporters for the hydrolysis probes analysed in multiplexed reactions between the 5 genes (CDKN1A, FXDR, SESN1 and PCNA in combination with the housekeeping gene HPRT1) per run. Table 2 below displays the oligonucleotide primer and probe sequences. Cycling parameters were 2 min at 95 °C, then 45 cycles of 10 s at 95 °C and 60 s at 60 °C. Data was analysed by the Rotor-Gene Q Series Software. Gene target Ct (cycle threshold) values were normalized to a Hypoxanthine-Guanine phosphoribosyl transferase 1 (HPRT1) housekeeping control. Ct values converted to transcript quantity by using standard curves that obtained by serial dilution of the PCR-amplified DNA fragments of each gene. The linear dynamic range of the standard curves covered six orders of magnitude (serial dilution from 3.2 × 10^−4^ to 8.2 × 10^−10^) producing PCR efficiencies between 91% and 103% for each gene and an efficiency score of R2 > 0.998. The log2 (Fold change) in gene expression levels are presented within, based on the average gene expression normalized to the housekeeping gene HPRT1 and 0Gy non-irradiated control.

### 4.6. Statistical Analysis

All statistical analysis were performed on Graphpad PRISM 9.1.2 (226). The non-parametric statistical tests such as the Mann–Whitney test compared G2 mean data sets and the Kruskal–Wallis tested the significance of the relative gene expression. Data sets in matched donor samples per IR dose (0.05 Gy and 0.05 Gy) were compared through the Wilcoxon Signed Ranks test.

## 5. Conclusions

In short, the conventional G2 chromosomal radiosensitivity assay bridged with genetic components of the DNA damage and repair pathway, in the form of a 4-gene signature panel of CDKN1A, FDXR, SESN1 and PCNA demonstrated good radiation-induced response for low doses of 0.05 Gy and 0.5 Gy IR and could be used as a potential biomarker in clinical settings. However, it is clear from intra-individual heterogeneity evident in these biomarkers that the radiosensitive phenotype is very complex, but it is anticipated that molecular advances will help to explore the radiosensitivity genotype further for biomarker discovery.

## Figures and Tables

**Figure 1 ijms-22-10607-f001:**
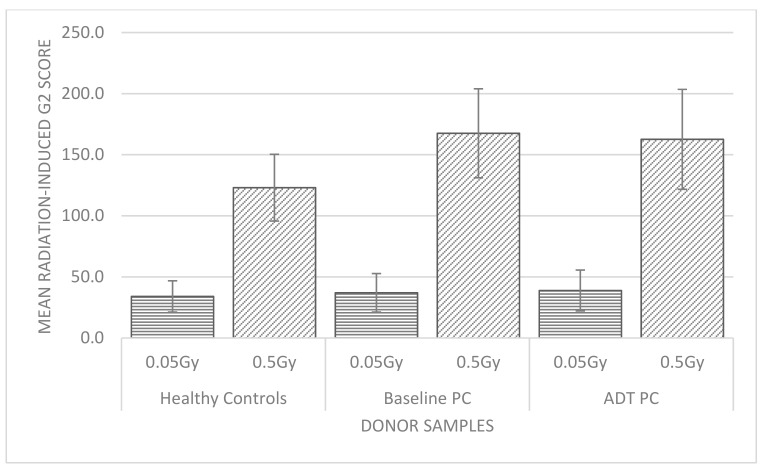
Mean radiation-induced G2 chromosomal radiosensitivity scores for each donor sample group at 0.05 Gy and 0.5 Gy IR. G2 chromosomal radiosensitivity scores were calculated as the number of aberrations per 100 metaphases for all donor samples at 0, 0.05 and 0.5 Gy and adjusted according to the 0 Gy G2 score for radiation-induced G2 scores for both 0.05 Gy and 0.5 Gy doses per donor sample.

**Figure 2 ijms-22-10607-f002:**
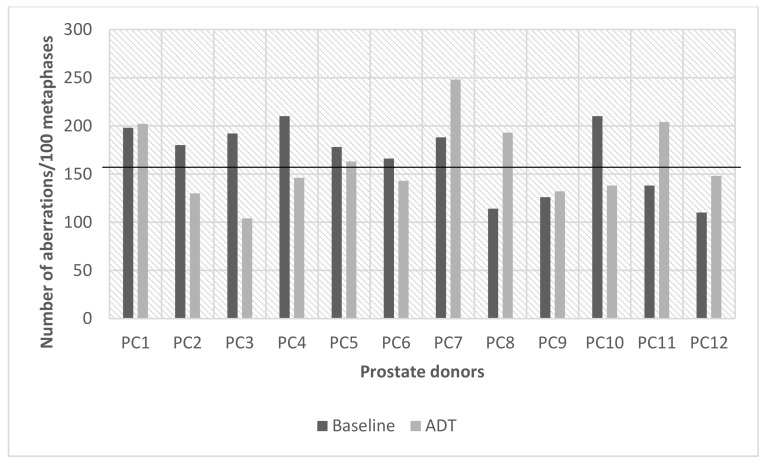
Radiation-induced G2 chromosomal radiosensitivity scores for each prostate cancer donor at baseline compared to ADT for 0.5 Gy IR. The 90th percentile of 152 aberrations/100 metaphases is also displayed as black line traversing all prostate donors.

**Figure 3 ijms-22-10607-f003:**
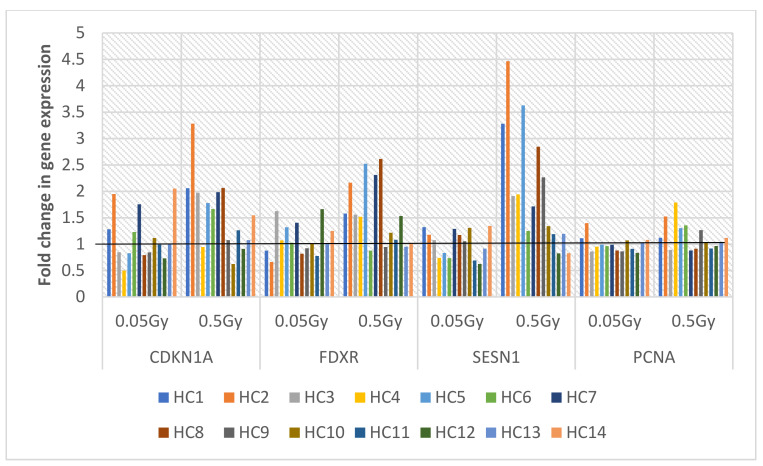
MQRT-PCR of 4 target genes CDKN1A, FDXR, SESN1 and PCNA for 14 healthy control donor samples exposed to 0.05 Gy and 0.5 Gy. The fold change threshold is displayed at 1 with a black line traversing each sample.

**Figure 4 ijms-22-10607-f004:**
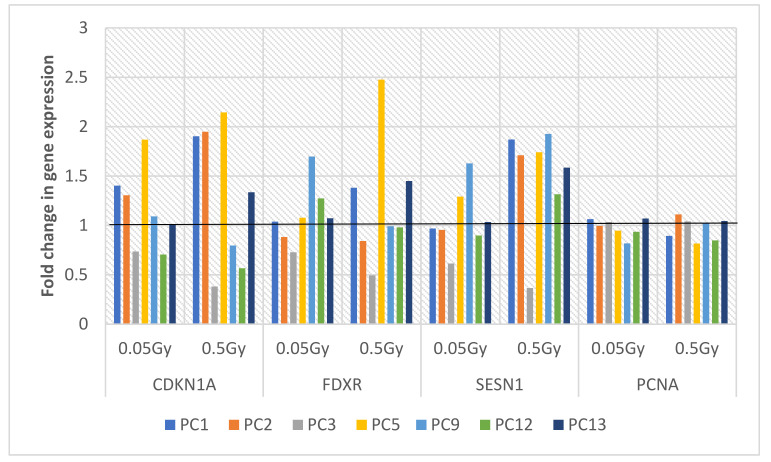
MQRT-PCR of 4 target genes CDKN1A, FDXR, SESN1 and PCNA for 5 baseline prostate cancer donor samples exposed to 0.05 Gy and 0.5 Gy. The fold change threshold is displayed at 1 with a black line traversing each sample.

**Figure 5 ijms-22-10607-f005:**
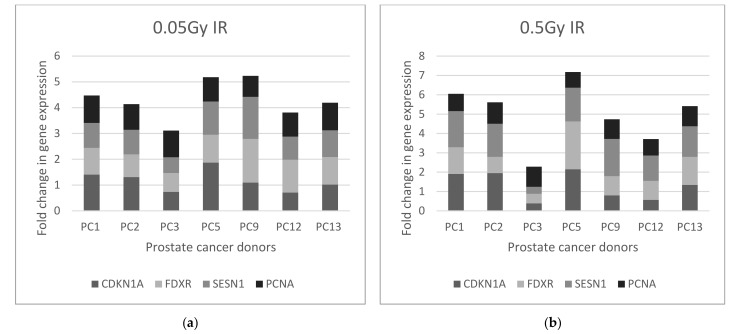
MQRTPCR fold change in gene expression for 4 target genes CDKN1A, FDXR, SESN1 and PCNA in prostate cancer donor samples taken at (**a**) Baseline exposed to 0.05 Gy, (**b**) Baseline exposed to 0.5 Gy; (**c**) ADT exposed to 0.05 Gy and (**d**) ADT exposed to 0.5 Gy. Baseline prostate donor samples coded PC (*n* = 7) and ADT prostate donor samples code PCT (*n* = 3).

**Table 1 ijms-22-10607-t001:** Radiation-induced G2 chromsomal radiosensitivity scores of the healthy control donor cohort (*n* = 45) at 0.05 Gy and 0.5 Gy IR doses.

Statistic	0.05 Gy	0.5 Gy
Mean	33.7 *	122.6 *
Standard deviation	12.7	27.3
Coefficient of variation	37.9	22.3
90th percentile	50 *	152 *

* Values represented as number X of aberrations per 100 metaphases (X abs/100 meta).

**Table 2 ijms-22-10607-t002:** Oligonucleotide primers and probes used for MQRT-PCR analysis. Primers designed by colleagues at Public Health England for biodosimetry studies [13,14].

Gene	Primers (5′-3′)	Probes (5′-3′)
*** HPRT1**	**F**-TCAGGCAGTATAATCCAAAGATGG	CGCAAGCTTGCTGGTGAAAAGGACCC
**CDKN1A**	**R**-AGTCTGGCTTATATCCAACACTTCGT **F**-GCAGACCAGCATGACAG **R**-TAGGGCTTCCTCTTGGA	TTTCTACCACTCCAAACGCCGGCT
**FDXR**	**F**-GTACAACGGGCTTCCTGAGA **R**-CTCAGGTGGGGTCAGTAGGA	CGGGCCACGTCCAGAGCCA
**SESN1**	**F**-GCTGTCTTGTGCATTACTTGTG **R**-CTGCGCAGCAGTCTACAG	ACATGTCCCACAACTTTGGTGCTGG
**PCNA**	**F**-CTCAAGGACCTCATCAACGA **R**-GGACATACTGGTGAGGTTCA	CCGCTGCGACCGCAACCTGG

* HPRT1 is the reference gene for the study against the 4 target genes

## Data Availability

The data presented in this study are available on request from the corresponding author. The data are not publicly available due to the use of individual donor blood samples in the study.

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
