# Peer review of "A 4-Gene Signature of CDKN1, FDXR, SESN1 and PCNA Radiation Biomarkers for Prediction of Patient Radiosensitivity"

_ijms, 2021, doi:10.3390/ijms221910607_

Round 1

Reviewer 1 Report

This is a very nice and concise paper dealing with individual response to radiation. The biomarkers are expression of 4 genes of interest and G2 assays and the report is devoted to prostate cancer patients. The quality of this paper is good. However, there are some major concerns that require simple but necessary rephrasing : 

The G2 assay (historically called G2 radiosensitivity assay) is not per se a predictive assay for radiosensitivity (as predisposition of post-radiotherapy adverse tissue reactions) but  rather radiosusceptibility (predisposition of radiation-induced cancer) and spontaneous cancer. Besides, the choice of the doses (0.05 and 0.5 Gy) is not relevant at all to predict post-radiotherapy reactions. Furthermore, the reaction severity grades for prostate patients are very low (1 or 2 instead of 3 or 4). Hence, this study focuses more on non cancer-cancer patients rather than clinical radiosensitivity among cancer patients. Hence, while the molecular data are interesting and useful, the clinical characterization of patients is made confused by the use of the radiosensitivity term : clinicians cannot consider prostate cancer patients with grades 1 or 2 as radiosensitive patients.

Instead of providing new data at 2 Gy, this reviewer proposes the following changes:

  • in Materials and Methods, the choice of the dose must be justified clearly by using molecular arguments rather than by using clinical ones (again, 0.05 and 0.5 Gy do not correspond to anything for prostate cancer patients)
  • in Results or in Supplementary data, the authors should provide correlations or absence of correlation between molecular endpoints ( G2 assay data and gene expression) in one side and severity grades on the other side. If there is no correlation, it should be discussed that the molecular endpoints can discriminate cancer and non cancer patients but cannot discriminate grade 2 and grade 1 cancer patients with the doses used. Again, the genes studied here are not necessarily responsible for the highest radiosensitivities when mutated. Conversely, they are associated with cancer proneness.
  • Mathematically, the discriminating power of a given population is always higher with 4-parameter (here, genes) than with 3-parameter or 2-parameter screening test. It would be therefore interesting to show the increase of the statistical robustness by using indifferently 2, 3 or 4 gene screening 
  • in Discussion, some paragraph should be added or some rephrased by introducing the notion that the results obtained provide interesting clues about cancer proneness rather than clinical radiosensitivity.

Author Response

Please see response to reviewer 1 attached.

Reviewer 2 Report

The authors have made great efforts to develop new assays for evaluation of radiation sensitivity. For this purpose, they evaluate expression level of  four radiation-responsive genes (i.e. CDKN1, FXDR, SESN1 and PCNA1) as a new assay, compared with classical chromosomal aberration assay against healthy control donor and prostate cancer donor at 0.05 Gy and 0.5 Gy doses. 

They  described this article very  carefully, so I can understand why and how they picked up above four genes for their purpose, and what they have done. But I cannot reach a conclusion whether this 4-gene expression assay is suitable for their purpose or not. (For example, if you evaluate 4-genes MQRTPCR  assay for healthy control donors like Figure 5, how it looks.)

Author Response

Please see attached file for response to reviewer 2.

Round 2

Reviewer 2 Report

I carefully read this revised manuscript and letter, and found my misunderstanding points. Now, I understand and agree with  authors’ points.  I think this revised manuscript is written in good faith, and enough to publish.  Therefore, I strongly recommend to publish this revised article to "International Journal of Molecular Sciences".

This manuscript is a resubmission of an earlier submission. The following is a list of the peer review reports and author responses from that submission.

Round 1

Reviewer 1 Report

Search for specific radiosensitivity biomarkers in radiotherapy patients is an important task to progress towards personalised medicine. In this paper authors have described the relationship between tumour response to radiation and expression of specific proteins. A new 4-gene signature panel of CDKN1, FDXR, SESN1 and PCNA was examined using cytogenetic methods to determine radiosensitivity of patients.  The data presented or CDKN1, FXDR and SESN1 (but not PCNA) provide the evidence that IR-induced DNA damage response gene expression is significant for detection of radiation sensitive patients. 

Author Response

Dear reviewer,

Many thanks for your kind review. It is much appreciated!!

With regards,

Orla Howe

Reviewer 2 Report

Sir, 
I have reviewed the manuscript "A 4-gene signature of CDKN1, FDXR, SESN1 and PCNA radiation biomarkers to predict patient radiosensitivity" submitted by Orla Howe and co-workers to IJMS. 

The story is well-worded. It is placing emphasis on fundamental aspects of predictive radiation oncology and a multidisciplinary approach to the therapy of patients. It pledges for assays for individual radiation sensitivity and susceptibility assessment prior to therapy. However, to cut the critical review short, this manuscript summarises observation of intra-individual heterogeneity to radiation response (I quote line 188 written by the authors). I must sadly agree. 

Figures 1 and 2 very well illustrate the failure of search for any statistically significant outcomes. As I see, the radiation induces a significant response in individuals (which is predictable), but the extent and pattern are too individual to reach statistically convincing data across (even very small) populations. 

To the research organisation and statistics: I am not keen to approve the reasoning of a "selection of 14 healthy controls" (line 200) because other samples were not available for some experiments. The authors indicated in the abstract that the robustness of the study should be guaranteed by a much larger control group (healthy control (n=45) - see line 26).

Similarly, the prostate cancer (n=14) donor cohort (see line 26)  shrikes somewhere to 7  - meaning 3+4 respectively (line 218). Again, this is a somewhat messy setting for any reasonable statistical evaluation. 

In this light, the authors later suggested (line 391) that "multiple samples per donor if practicable" should be collected. It is a jocular recommendation. 

A very great problem is the (almost missing)  population description. The authors presented age - however, the cohorts are mostly not overlapping (normal age ranges 21-64 years), and the cancer patients cohort is much older (60-85 years old). As the authors decided to analyze only segments of their initial cohorts, this difference can be even greater.  I must point out that age inequity is a critical problem in this manuscript. Ageing is an important risk factor for cancer - the authors could very easily find this topic extensively reviewed in the last years in the medical literature. The authors should acknowledge this. The gene repair machinery is significantly failing even in healthy individuals who have finished their reproductive age. The authors are dealing with hormonal deprivation; however, we actually do not have any insight into the endocrine status of their patients. This is a critically important aspect related to gene repair and potential also to the radiotherapy effect. 

The authors have decided to use a very conventional approach and tested "general radiosensitivity". Blood samples are a convenient and traditional approach - but it is a surrogate. The peripheral blood test is not necessarily a reliable predictor of tumour radiosensitivity or real clinical outcome. This blood testing can illustrate the frailty of the elderly population. 

Minor points: 

Section 1 is nicely written for a review article, but it seems too wordy compared to the very modest outcomes of this research paper. 

Wilcoxin signed ranks test. ---should be Wilcoxon (line 141,165, 185, 286). 

To conclude, I must agree that the authors have done some interesting experiments and they also gained some interesting data. Regrettably, I can't entirely agree with the data handling (sometimes missing) description and interpretation. However, this might be changed, and additional experiments should be done if necessary. At this stage, I would recommend the rejection of the manuscript. I would also encourage the authors to reconsider their manuscript thoroughly and submit a modified version soon to IJMS again. I am very keen to review it again. 

Round 2

Reviewer 2 Report

Sir,

I have recently reviewed the revised version of the manuscript "A 4-gene signature of CDKN1, FDXR, SESN1 and PCNA radiation biomarkers for prediction of patient radiosensitivity" submitted by Orla Howe and co-workers to IJMS. I have also studied the rebuttal letter with utmost care. 

I am very glad to see that the authors corrected the spelling mistake and added one reference which was requested by me. I am grateful for that. 

However, I must regrettably say that the manuscript is not otherwise substantially changed.

The authors also provided a supplementary table summarizing prostate cancer patients 1-12. Surprisingly, Figure 5 also contains PC13 and PCT18 (and also Figure 4). Who are these patients? The identification of patients in the body of the manuscript and the list in the supplementary table is obviously not matching well. 

Further, the authors declared a mistake in their original dataset, so the youngest prostate cancer patient is now 57 y.o.   However, we do not know anything about the age distribution of the 14 men of the control group (The rebuttal letter says: the donor control samples were derived from researchers and staff at the institute (pre-retirement age). The article still says "21-64 years old". Further, the authors claim (quote): "50% of the prostate cancer donors were 57-65years, which does cross over the age range of the health control donors". Generally, I am afraid that this is a rather vague description. What does this mean statistically? The authors are presenting very small populations. One must be extremely careful when dealing with these small populations and the control group must simply match perfectly. 50% seems to be a somewhat weak basis. 

To conclude,  I must regrettably say that I do not see in this version sufficient improvement to change my earlier recommendation to reject the manuscript.